# The Effect of Feeding with Central European Local Mulberry Genotypes on the Development and Health Status of Silkworms and Quality Parameters of Raw Silk

**DOI:** 10.3390/insects13090836

**Published:** 2022-09-14

**Authors:** Andreja Urbanek Krajnc, Tamas Bakonyi, Istvan Ando, Eva Kurucz, Norbert Solymosi, Paula Pongrac, Rebeka Lucijana Berčič

**Affiliations:** 1Faculty of Agriculture and Life Sciences, University of Maribor, Pivola 10, 2311 Hoče, Slovenia; 2Department of Microbiology and Infectious Diseases, University of Veterinary Medicine Budapest, István u. 2, H-1078 Budapest, Hungary; 3Institute of Genetics, Biological Research Center of the Hungarian Academy of Sciences, P.O. Box 521, H-6701 Szeged, Hungary; 4Centre for Bioinformatics, University of Veterinary Medicine Budapest, István u. 2, H-1078 Budapest, Hungary; 5Department of Biology, Chair of Botany and Plant Physiology, Biotechnical Faculty, University of Ljubljana, Jamnikarjeva 101, 1000 Ljubljana, Slovenia; 6Jožef Stefan Institute, Jamova 39, 1000 Ljubljana, Slovenia; 7Institute for Sericulture, Rebecca Luciana Bercic, Koroška c. 65, 2000 Maribor, Slovenia

**Keywords:** silkworm, mulberry leaves, chemical composition, feeding experiment, bacterial microbiome, BmNPV

## Abstract

**Simple Summary:**

Several regions of Slovenia and Hungary retained numerous centuries-old white mulberry trees, evidence of past sericultural activities, being traditionally used to feed the silkworm larvae. Attempts for the reintroduction of sericulture in these countries are ongoing. The current study assessed the suitability of the locally adapted mulberry trees for contemporary sericultural needs. Silkworm hybrids were fed with leaves of the selected local mulberry genotypes and the larvae performance parameters (bodyweight, spinning success, cocoon quantity, and quality) were compared to those fed with reference mulberry varieties. The chemical contents and nutritive parameters of the mulberry leaves were determined, and connections were predicted between selected leaf compounds and silkworm performance parameters. The local mulberries had higher total protein contents, and lower total phenolic contents and differed in some individual phenolics, macro- and microelements compared to the reference sericultural and fruit varieties. A combined positive influence of proteins, specific phenolics, and microelements on larval growth and silk thread parameters was predicted. The health status and gut microbiome compositions of larvae were also analyzed. The results of the study indicate that selected local Slovenian and Hungarian mulberry varieties are suitable for high-quality silk cocoon and raw silk production.

**Abstract:**

Silkworm rearing activities ceased in the 1970′s in several European countries. Attempts on the re-establishment of ecological and sustainable sericulture in Slovenia and Hungary are ongoing. The aim of the study was to assess the usability of locally adapted mulberry genotypes for sericulture and to estimate connections between leaf compound and silkworm performance parameters. A controlled feeding experiment of silkworms was performed to test the influence of leaves from selected trees on the growth of larvae, the health and microbiological status of larvae (e.g., gut bacterial microbiome, *Bombyx mori nucleopolyhedrovirus* infection), weight of cocoons and raw silk parameters. The Slovenian and Hungarian mulberry genotypes had significantly higher total protein contents, and lower total phenolic contents and differed significantly in some individual phenolics compared to the reference sericultural and fruit varieties. Significant differences were found in the contents of the macro- and microelements, namely S, Mn, Fe, and Sr. Based on correlative statistics and multivariate analysis, a combined positive influence of proteins, specific phenolics, and microelements on larval growth and silk thread parameters was predicted. The results of the study indicate that selected local Slovenian and Hungarian mulberry varieties are suitable for high-quality silk cocoon and raw silk production.

## 1. Introduction

Although China and India are the current monopolists in silk production, representing more than 97% of the global sourcing of this natural fiber, there are initiatives for the re-establishment of silk production in European countries which were major silk producers in the past centuries [1]. In the time of the Austro-Hungarian Empire, sericulture was highly developed in the territories of today’s Slovenia and Hungary. Due to the competition in the global silk market, highly developed silkworm rearing in these two countries ceased approximately 60 years ago [2,3,4]. Nowadays, all around these countries, we can still see up to 600 years old white mulberry (*Morus alba* L.) trees—living monuments of sericultural history, as the leaves of these trees were used for feeding the larvae of the domestic silk moth (*Bombyx mori* L.) over centuries. These mulberry genotypes are adapted to the local climatic and environmental conditions and were appropriate for feeding silkworm larvae in the past. Sericulture subsided in the previous decades in Italy, and nowadays production is increasing [5]. In Italy currently, high-performance hybrids of *B. mori* are used for silk production, which are typically fed by selected mulberry genotypes. One of the most important practical questions of the sericulture re-establishment attempt is, whether the locally adapted genotypes in other European countries are appropriate for feeding the modern hybrids of silkworm, or the reference varieties are necessary to achieve appropriate performance (i.e., quality of silk cocoons according to the standards and market’s expectations, and quantities comparable with rearing farms in Italy).

Recently, a joint Slovenian-Hungarian basic research project was implemented to define the conditions for the re-establishment of ecologically and economically sustainable sericulture. The main research question was whether the development and health status of silkworm larvae, production of silk cocoons, and quality of raw silk are affected by the diversity and nutritive characteristics of locally adapted mulberry trees native in Hungary and Slovenia compared to reference varieties currently used for silkworm rearing. As a multidisciplinary research approach, local mulberry genetic resources were catalogued and their biochemical characteristics were analysed [6,7]. The aim of the presented research was to test the selected genotypes in a silkworm feeding experiment and to compare their effect on silkworm development, cocoon production, the health status of silkworm larvae, and raw silk parameters, compared to reference mulberry varieties.

## 2. Materials and Methods

### 2.1. Mulberry Material for Silkworm Feeding Experiment

Mulberry genotypes from the mulberry collection were chosen for the feeding experiment based on previous screening of metabolites in leaves [6,7], and were organized into three sections: the first part is represented by Italian sericultural *M. alba* varieties (‘Giazzola’, ‘Florio’, ‘Morettiana’) and the Japanese variety ‘Kokusou-20’ obtained from the gene bank at the Centro di Ricerca per l’Agricoltura e Ambiente (CREA-AA), Laboratorio di Gelsibachicoltura di Padova, Italy. The second part comprises vegetatively propagated trees derived from the local historical Slovenian and Hungarian mulberries, which were obtained during the classification of the mulberry gene pool. The third part of the collection is intended for growing recent varieties of *M. alba*, *M. nigra* and *M. australis* as well as hybrids of *M. alba* × *rubra* suitable for fruit production.

The choice of mulberry trees for reference plantation feeding experiments was conducted according to: (1) geographical distribution of the original old local mulberry genotypes in Hungary and Slovenia; (2) multivariate analyses of previous biochemical values of original trees that allowed us to define seven chemotypes in more details in terms of the composition of individual amino acids and phenolics [6,7], out of which genotypes rich in total proteins, total phenolics, chlorogenic acid, and particular flavonoids were chosen. For the experiment, three to five trees of the same genotype were harvested.

Mulberry trees are grouped into four main sections:(A)Slovenian mulberry genotypes: selected, locally adapted genotypes, obtained by cuttings from old (trunk diameter > 180 cm) Slovenian *M. alba* trees (*n* = 10);(B)Hungarian mulberry genotypes: selected, locally adapted genotypes, obtained by cuttings from old (trunk diameter > 180 cm) Hungarian *M. alba* trees (*n* = 16);(C)reference sericultural *M. alba* varieties, obtained from the mulberry gene bank of the CREA-AA (‘Kokusou-20’, ‘Morettiana’, ‘Florio’, ‘Giazzola’);(D)varieties of *M. alba*, *M. alba* × *rubra*, *M. nigra* and *M. australis* grown for fruit production.

The list of mulberry genotypes along with their location and geographic coordinates of historical mulberry trees are presented in Appendix A.

### 2.2. Determination of the Chemical Composition of Mulberry Leaves

For biochemical analyses, five to seven fully developed sun-exposed leaves (5th to 7th leaf below the apex) from a one-year branch were collected randomly per each sampled tree (3–5 trees per selected genotype) and used as one sample. The trees were sampled twice during the experiment on 16th and 23rd of June 2021. Samples were immediately stored on dry ice, later transferred to a freezer at −80 °C, and subsequently freeze-dried and grounded. The prepared samples were then stored in airtight vials at −20 °C prior to biochemical analyses. The concentrations of the analyzed nutrients are calculated on a dry weight (DW) basis.

Total proteins were determined spectrophotometrically at 595 nm, following the Bradford procedure [8]. The total protein content was calculated based on a standard curve, which was prepared using bovine serum albumin (BSA, 0.05–0.25 mg mL^−1^) and expressed as mg BSA equivalent per 100 g of dried mulberry leaves (mg BSA/100 g DW).

Total phenolics of the methanolic extracts (3% formic acid in 95% methanol) were determined using the Folin-Ciocalteu method following the procedure of Ainsworth and Gillespie [9]. The absorbance was measured at 765 nm against a reagent blank (3% formic acid in 95% methanol). Gallic acid was used as the reference standard (GA, 0.025–0.25 mg mL^−1^). The total PH was expressed as mg gallic acid equivalents per 100 g of dried leaf sample (mg GAE/100 g DW).

Furthermore, the methanolic extracts were subjected to the gradient HPLC analysis as described in detail in a previous study [6]. All considered phenolic compounds were previously identified by a mass spectrometer (Thermo Finigan, San Jose, CA, USA) with an electrospray interface (ESI) operating in negative ion mode as previously described in Šelih et al. [7]. The identification of compounds was confirmed by comparing their spectra, retention times, and fragmentation as well as by adding the standard solution to the sample. Quantification was achieved by comparing with corresponding external standards (chlorogenic acid, kaempferol, p-coumaric acid, rutin, and quercetin; all obtained from Sigma Aldrich) of known concentrations. For the compounds for which the standards were not available, related compounds were used as standards. Therefore, quercetin-3-O-glucoside (isoquercetin), quercetin dirhamnosylhexoside and quercetin malonylhexoside were quantified in quercetin equivalents, kaempferol acetylhexoside in kaempferol equivalents, caffeoylquinic acid derivatives in equivalent of chlorogenic acid and p-coumaric and p-coumaroylquinic acid derivatives in equivalent of p-coumaric acid. The contents of individual compounds were expressed in mg/g DW.

Macro- and microelement analyses (phosphorus (P), sulphur (S), chlorine (Cl), potassium (K), calcium (Ca), manganese (Mn), iron (Fe), nickel (Ni), zinc (Zn), rubidium (Rb) and strontium (Sr)) were performed with a tabletop X-ray fluorescence (XRF) spectrometer PEDUZO T02 (Jožef Stefan Institute, Slovenia) with rhodium-anode X-ray tube. X-ray fluorescence was detected by a silicon drift diode detector (Amptek Inc., Bedford, MA, USA). The energy resolution of the spectrometer at count rates below 1000 cps was 140 eV at 5.9 keV. Measurements were performed in air and the samples were irradiated for 1000–5000 s to ensure sufficiently low statistical error [10,11]. The spectra were analysed by software operating in LabVIEW [12]. Element analysis was validated using standard reference material NIST SRM 1573a (tomato leaves).

### 2.3. Silkworm Rearing

#### 2.3.1. Silkworm Genetic Material

Certified, traceable silkworm genetic material (eggs of a polyhybrid strain) was obtained from CREA-AA, Laboratory of Sericulture in Padua, Italy. The polyhybrid (4-way-hybrid) strain was generated by crossing a parental Chinese strain (SC2 × SC3) × a parental Japanese strain (SG1 × SG3). Eggs proved to be negative to the microscopic pebrine analysis, which is routinely carried out to detect *Nosema bombycis* infection.

#### 2.3.2. Silkworm Rearing Technology

Estimation of the optimal timing for the start of silkworm rearing was determined by observation of the vegetative status of mulberry trees according to the reference guidelines of Brion [13].

The eggs of silkworm hybrids were transported from CREA-AA to the Faculty of Agriculture and Life Sciences, University of Maribor, Slovenia in controlled temperature conditions (25 °C). The eggs were weighed to have similar quantities for each lot, i.e., about 5000 larvae per thesis. In all five instars of silkworm development silkworms were reared in faculties facility near reference plantation in cardboard boxes, changed after each moult. Photoperiod was 12 h day-night. Rearing temperature and relative humidity all the time varied between 23–26 °C and 45–60%, respectively. Rearing place was constantly aerated.

Until the beginning of the 4th developmental instar larvae were reared together in a common box and fed with the mixture of leaves of Slovenian (Slo), Hungarian (Hu) mulberry genotypes and reference sericultural varieties (ref) of *Morus alba* trees from the mulberry collection (groups 1–4). After the 3rd moult larvae were randomly distributed to 38 cardboard boxes (25 × 55 × 5 cm) in groups of 30 larvae to test individual mulberry genotypes. For studies on gut microbiome analysis and haemocyte parameters, groups of larvae were fed with mixtures of leaves according to the four main groups (i.e., Slo, Hu, ref, and fruit varieties).

Larvae were fed ad libitum with leaves of selected genotypes/species of mulberry trees. When in the rearing box about 10% of leaves from the previous feeding were left, a new portion of food was added every 3–4 h during the day and every 6 h, during the night. Leaves were offered until the last larva started to spin or for 12 days after the last moult to those larvae which did not start to spin. After the start of each moult, food was not offered for two days, in order to obtain complete moulting from all the individuals of each lot and to start the following instar homogeneously.

The cumulative body weight of the larvae within one group was measured every 24 h. Larvae were observed for activities (e.g., eating, moulting, spinning) and health status (signs of any disease).

Cocoons were collected eight days after beginning of spinning. The number and cumulative weight of the cocoons were measured in each experimental group. Cocoons were dried to 42–45% of the original fresh weight at 60 °C.

#### 2.3.3. Monitoring the Health and Microbiological Status of Silkworm Larvae

The activity of larvae was checked at least six times per day, during feeding and bodyweight measuring. Larvae showing signs of illness (e.g., inactivity, cessation of eating, sluggishness, or flaccidity, colour changes, swollen body, fragile intersegmental membranes, diarrhoea) or dead larvae were immediately removed from the group and were stored at −20 °C until processing.

##### Molecular Detection of Bombyx Mori Nucleopolyhedrovirus

*Bombyx mori nucleopolyhedrovirus* (BmNPV, *Baculoviridae*) (the causative agent of Grasserie disease) has previously been detected in larvae at CREA-AA [14]. Vertical (transovarial) transmission occurs with BmNPV, and the virus can persist in some larvae as sublethal infection [15,16]. Therefore, diseased larvae were tested for infection by in-house developed, qualitative real-time PCR (rt-PCR) assay using the TaqMan technology. Larval specimens were homogenised in sterile ceramic mortars, and 10× volume sterile phosphate-buffered saline (PBS) was added. Homogenates were centrifuged at 1500× *g* for five minutes and total DNA was extracted from 200 μL supernatants, using the Qiagen Viral DNA extraction kit (Qiagen, Hilden, Germany). The rt-PCR assay was targeting the polymerase gene region (genomic primer: 5′-GCCACCGTAATCACRCGTCTTT-3′, complementary primer: 5′-CGATAACCCGGGCAAAAA-3′ and TaqMan probe: 5′-FAM-ACCTTCATTATTATCGTCAGCCGATTTGCG-TAMRA-3′. Samples were tested in duplicates. The specificity of the assay was confirmed by direct sequencing of the amplification products and identification through BLAST search in the gene bank databases. Relative quantification of BmNPV DNA loads in the samples was based on cT values.

##### Identification of Gut Microbiome and Potential Bacterial Pathogens by Metagenomic Studies

Three randomly selected, healthy 5th instar larvae were collected from groups fed with Slovenian and Hungarian mulberry genotypes, reference varieties, *M. australis*, and *M. nigra*. The guts of larvae from each pool were removed, homogenized, and total DNA was extracted. The DNA samples were submitted to a metagenomic investigation targeting the 16S rRNA gene of bacteria to reveal the bacterial gut microbiomes of silkworms, fed with different genotypes and species of mulberry leaves.

After merging the paired-end reads by PEAR quality-based filtering, trimming was performed by Trimmomatic, using 20 as the quality threshold, and reads longer than 50 bp were retained only [17,18]. The remaining reads after deduplication by VSEARCH [19] were taxonomically classified using Kraken2 (k = 35) with the NCBI non-redundant nucleotide database for the shotgun sequenced samples [20,21]. In the case of 16S rRNA sequenced samples for the classification the Greengenes database was used following chimera filtering by VSEARCH [19,22]. The taxon classification data was managed in R (R Core Team) using functions of package phyloseq [23,24].

##### Comparison of Selected Qualitative and Quantitative Hemogram Parameters in the Main Feeding Groups of Larvae

For the description of the immune cells, 10 fifth instar larvae were collected from different experimental groups. Haemolymph was collected by cutting a proleg into Schneider’s insect medium supplemented with 5% foetal bovine serum and phenylthiourea. Tenfold dilution of the haemolymph in the same medium was used to determine total cell count in a Bürker counting chamber and 30 μL aliquots of the diluted cell suspensions were placed on the spots of Hendley Essex—Diagnostic Microscope Slides for phenotype analysis. The haemocytes were allowed to settle and adhere to the slide for 60 min in a humidity chamber. After the one-hour incubation, the haemocytes were fixed with acetone for six minutes and were classified according to morphological criteria [25,26]. The proportions of three main haemocyte classes, the granular cells, the plasmacytes, and the prohemocytes were determined.

### 2.4. Methods for Evaluation of Quality of Silk Cocoons

The number and cumulative weight of the cocoons were measured in each experimental group. Cocoons were dried until they reached approximately 45% of their original weight at 60 °C and sent to CREA-AA, Laboratory of Sericulture, for the analysis.

The whole cocoon weight, the silk shell weight, and the silk percentage were calculated, after cutting cocoons and extracting pupae on around 20 cocoons per sample. The silk percentage was calculated according to Lee [27].

Ten cocoons for each experimental sample were used for experimental reeling on a testing reeling machine apt to reel individual cocoons, by recording the silk thread length, the title in deniers, the number of breakages, and the wastes.

### 2.5. Statistical Data Analysis

The results of biochemical analyses of mulberry leaves are shown as mean (average) values (±standard deviation, SD) of the analyses on two sampling dates (16th and 23rd of June 2021) during the feeding experiments, in which three to five trees were sampled and used as one sample. Measurements were performed at least four times for each sample and in duplicate. Assumptions of normality for all chemical traits were checked with Kolmogorov–Smirnov test.

The chemical traits of mulberry leaves, silkworm weight (5–7th day/5th instar), cocoon weight (fresh), silk thread parameters, and reeling wastes were presented by means and SD and were statistically evaluated by one-way analysis of variance (ANOVA), followed by post-hoc comparison according to Duncan. Letters describe significant differences among genotypes and origin-dependent genotype groups.

The Pearson correlation coefficient (Sig. 2-tailed) was calculated between evaluated chemical parameters, silkworm weight, cocoon weight, and silk thread parameters, to analyze the correlative relationship between the measured parameters and to find out the most effective differentiating traits.

Principal component analysis (PCA) enabled us to perform a comprehensive assessment of chemical traits of individual trees, silkworm weight, cocoon weight, and silk thread parameters by discriminating geographical distribution with respect to reference varieties.

IBM SPSS Statistics 25 (Armonk, New York, NY, USA; 2017), StatSoft, Inc. Statistica 8.0 (Victoria, Australia; 2007) and Past 3.17 (Zürich, Switzerland; 2020) software were used for statistical analysis [28].

## 3. Results

### 3.1. Chemical Composition of the Leaves of Selected Slovenian and Hungarian Mulberry Genotypes, Reference Sericultural and Fruit Varieties

The results of biochemical analyses of the total proteins, total phenolics and individual phenolic components in leaves are presented in Figure 1A,B and Appendix A. The highest total protein content was determined in genotype SM 101.1 (239.42 mg/g DW), the lowest in the Bulgarian fruit genotype. Phenolics were the highest in *M. alba* × *rubra* (21.51 mg/g DW) and the lowest in Slovenian genotype SM 6 (11.26 mg/g DW) (Figure 1A, Appendix A).

By evaluating the individual phenolics in mulberry leaves, eight different hydroxycinnamic acids and eleven flavonols were identified (Figure 1B, Appendix A). The main phenolic acids, from the hydroxycinnamic group were caffeoylquinic derivatives (with chlorogenic acid predominating) and p-coumaroylquinic acid derivatives. The predominant flavonoids were quercetin and kaempferol glycosides. The main quercetin glycosides were rutin, quercetin malonyl-hexoside, and quercetin-3-glucoside (isoquercetin), whereas the predominant kaempferol glycoside was kaempferol acetyl-hexoside (Figure 1B, Appendix A).

The maximum concentration of chlorogenic acid was determined in *M. nigra* (18.05 mg/g DW), the minimum concentration in Hungarian genotype SO 1042 (4.00 mg/g DW). Within the Slovenian mulberry genotypes, the highest concentration was determined in SE 5 (11.73 mg/g DW), whereas within the Hungarian genotypes in SO 1035 (13.52 mg/g DW). Among the reference varieties ‘Kokusou-20’ yielded the highest chlorogenic acid content (17.67 mg/g DW), whereas the concentration of 4-caffeoylquinic acid (4.36 mg/g DW) was the highest in *M. alba* × *rubra*. The above-mentioned genotypes superior in chlorogenic acid were also characterized by the highest total caffeoylquinic acid derivatives (Figure 1B, Appendix A).

The predominant coumaroylquinic acid derivative was 5-coumaroylquinic acid, whose concentration was the highest in *M. nigra*, yielding a total coumaroylquinic acid amount of 10 mg/g DW, followed by *M. alba* × *rubra* with 7.39 mg/g DW.

Among quercetin glycosides, rutin concentration was the highest in Hungarian genotype TO 1131. Maximum quercetin malonylhexoside was determined in ‘Kokusou-20’ (2.25 mg/g DW). Quercetin-3-glucoside and kaempherol acetyl-hexoside were the highest *M. alba* × *rubra*. This hybrid was also characterized by the highest total kaempherol glycoside derivatives (6.27 mg/g DW) (Figure 1B, Appendix A).

The results of the element analysis of mulberry leaves are shown in Figure 2. P concentrations ranged between 1.07 and 2.04 g/kg DW and were the highest in genotype SE 5. Some mulberries contained more than 1 g/kg DW of S with the highest concentration in genotype GMS 2532 (1.26 g/kg DW). K reached maximum concentration in genotype BA 2225 (22.1 g/kg DW) and Ca in genotype GMS 2532 (20 g/kg DW) (Figure 2A, Appendix A).

Among microelements, maximum Cl concentrations were found in genotype BE 1264.2 (1320 mg/kg DW). Fe and Ni concentrations were the highest in genotype GMS 2533. The highest Zn concentration was found in Slovenian genotype SM 6. ‘Kokusou-20’ was found to be rich in Rb (59.4 mg/kg DW), whereas Sr concentrations ranged from 22.8 to 152 mg/kg DW with the maximum value in GMS 2533 (Figure 2B, Appendix A).

### 3.2. Origin Dependent Differences in Chemical Composition of Mulberry Leaves

The mulberry genotypes were categorized according to their origin to Hungarian and Slovenian old mulberry genotypes and compared to the reference and fruit varieties. Based on the statistical evaluation of the chemical compounds we determined the significant highest amounts of proteins in both Slovenian and Hungarian genotypes of *M. alba*, when compared to fruit varieties, which had the lowest protein contents. The sericultural reference varieties were intermediate in proteins (Table 1, Appendix A).

The significant highest amount of total phenolics was observed in classical contemporary sericultural genotypes and fruit varieties. In Table 1 predominant individual phenolics are summarised as mean values of Slovenian, Hungarian local genotypes, reference sericultural, and fruit varieties, whereas the ranges of all analysed phenolics of individually selected mulberry genotypes are presented in Appendix A. The least mean content of chlorogenic acid was found in Slovenian genotypes, the most in fruit varieties. Similarly, the highest amount of 4-caffeoylquinic acid was found in fruit varieties. The highest amount of rutin was analysed in reference and Hungarian genotypes of *M. alba*, the lowest in fruit varieties. Reference sericultural genotypes and fruit varieties have the highest amount of quercetin-3-glucoside, the least concentration was determined in Slovenian genotypes of mulberry trees. The highest amount of quercetin malonyl-hexoside was found in the leaves of Hungarian and reference mulberry trees, the lowest in fruit varieties (Table 1, Appendix A).

Significant differences between the mulberry groups in the contents of the main elements were determined for S, Mn, Fe, and Sr. Slovenian mulberry genotypes were characterized by the highest S content, whereas the lowest was determined in fruit varieties. Both, Slovenian and Hungarian genotypes were rich in Mn content. Fe was significantly highest in Hungarian genotypes and lowest in fruit varieties. Sr was significantly enhanced in Hungarian genotypes, whereas the lowest concentrations were found in reference varieties (Table 2, Appendix A).

### 3.3. Description of the Influence of Feeding Silkworms with Leaves of Various Genotypes of Mulberry Trees on Larval Development

The highest silkworm weight was obtained when silkworms were fed with the Hungarian genotype BA 2151 and the lowest with ZA 1060 (Appendix A). Out of Slovenian varieties, the maximum weight was obtained when larvae were fed with leaves of SP12 mulberry genotype, the lowest weight was determined for larvae that were fed with Submediterrean varieties (SM6, SM101.1, SM137). Among the reference varieties, significantly higher weight was obtained by feeding them with ‘Florio’. Among the fruit varieties, larvae fed with *M. nigra* gave the highest weight (Figure 3, Appendix A).

### 3.4. Origin Dependent Differences in the Growth of the Silkworm Larvae and Cocoon Weight

Based on statistical evaluation of the four main groups (Slovenian, Hungarian, reference sericultural and fruit varieties) reference sericultural varieties were characterized by the highest larval growth, whereas no statistical differences were found between mean cocoon fresh weight (Table 3).

### 3.5. Description of the Influence of Feeding Silkworms with Leaves of Various Mulberry Genotypes on the Quality of Raw Silk

Among Slovenian varieties, the maximum length was obtained when larvae were fed with Slovenian variety SP 256. Superior length was also obtained with GMS 2286, whereas the lowest was obtained with the reference varieties ‘Giazzola’, Bulgarian fruit genotype, and *M. alba × rubra*, which also gave the lowest raw silk weight. Significant highest weight was obtained with Hungarian genotype SO 1013. The Hungarian genotype BE 1264.2 was characterized by the highest thickness and the Slovenian genotype SM 137 with the lowest. When analysing the reeling wastes, the highest struse was found in most of the Hungarian varieties and the significant highest telette was determined in Hungarian variety GMS 2533 (Appendix A).

### 3.6. Origin Dependent Differences in the Quality of Raw Silk

There was a trend for longer silk thread production when larvae were fed with Slovenian and Hungarian varieties in comparison to reference sericultural varieties, although the trend was insignificant. Fruit varieties were characterized by a shorter length and weight of raw silk. Struse was significantly largest when larvae were fed with Hungarian varieties, thickness and telette did not significantly differ among the groups. When analysing reeling wastes struse was highest in Hungarian genotypes and lowest in Slovenian genotypes and fruit varieties. There were no significant differences in telette (Table 4).

### 3.7. Correlation between Mulberry Metabolites, Silkworm, and Raw Silk Parameters

A high correlation was found between silkworm weight and raw silk thickness, as well as raw silk length, and weight; whereas a negative between silk thread parameters (length, weight) and reeling waste (telette, struse). The results of the chemical analysis of mulberry leaves indicate a strong correlation between total proteins and raw silk length and weight. Among phenolic acids, t-5-coumaroylquinic acid correlated strongly with the weight of silkworms. There was a medium negative correlation between p-coumaric acid hexoside and raw silk length. Among flavonols, rutin correlated positively with struse. There was a negative correlation between kaempherol-dirhamnosyl-hexoside and raw silk length and weight, whereas a positive correlation between kaempherol-rhamnosyl-hexoside and struse (Table 5).

There was no significant correlation between the macroelements and silkworm weight, silk thread parameters, and reeling waste. Among microelements, Cl correlated positively with silk weight, while Fe and Sr correlated with telette (Table 6).

### 3.8. Principal Component Analysis of Mulberry Metabolites, Silkworm, and Raw Silk Parameters

To enable a comprehensive assessment of the chemical composition of mulberry leaves, with respect to their genotype, larval growth and silk parameters, a principal component analysis (PCA) has been conducted. The discriminant function 1, which accounts for 78.14% of the variance, as explained by the model, was weighted most strongly by total phenolics, followed by rutin, quercetin glucoside, larval weight, and chlorogenic acid. It is further negatively associated with silk length, total proteins, Mn and Sr.

Thus, function 1 clearly (over 89%) separates reference varieties, which are characterized by higher larval weight, which coincides with high values of chlorogenic acid, rutin, and other quercetin glycosides, from Hungarian and Slovenian varieties, being higher in proteins, length, and weight of silk thread. Furthermore, it must be pointed out that some old Hungarian and Slovenian varieties coincide with high concentrations of Cl, Mn, Zn, and Sr.

The second discriminant function accounts for another 21.86% of the variance. It is positively associated with cocoon fresh and dry weight, length, total phenolics, Mn, and K and negatively with Ca and quercetin glucoside. Function 2 clearly separates Hungarian mulberry (by over 87%), which is characterized by higher cocoon fresh and dry weight and the above-mentioned elements from Slovenian genotypes being high in Ca (Figure 4, Appendix A).

Genotypes with the highest average cocoon fresh and dry weight are GMS 2532 and GMS 2329. Genotype TO1131 contained high total phenolic concentration, in particular quercetin-3-rutinoside which coincided with high 5th instar/5th day weight. BA 2225 is highest in Mn and K concentrations. Slovenian genotype A6 was high in Ca and certain quercetin glycosides as well as NG 214 in Ca content and caffeoylquinic acid glycosides (Figure 4, Table 1).

### 3.9. Health and Microbiological Status of Silkworm Larvae

All larvae were moulting synchronously, no prolonged instar duration was observed. Obvious unequal growth and development were observed by three larvae of various groups. During rearing healthy, but cachectic larvae were found only in groups fed with Hungarian (Hu) mulberry genotypes of trees GMS 2532 (fourth instar), SO 1042 (fourth instar), and TO 1131 (fifth instar).

Six ill or dead larvae (0.47% of all larvae) were found in the following developmental stages: 1st day of third instar (one dead larva); 2nd day of third instar (one ill cachectic larva); 1st day of fourth instar (one dead larva); 1st day of fifth instar (one dead larva in group Hu-BE 1264/2); end of fifth instar (one dead larva from group Hu-ZA 2084; one dead larva from group Hu-mix).

In the groups of larvae fed with Slovenian mulberry genotypes (*n* = 10) all larvae successfully spun in three groups. In five groups 5%, in one group 15% and in one group 20% of the larvae failed to spin. All larvae successfully spinned in the group fed with the mixture of Slovenian mulberry genotypes.

In the groups of larvae fed with Hungarian *M. alba* genotypes (*n* = 16) all larvae successfully spinned in seven groups. In another seven groups 5% and in two groups 15% of the larvae failed to spin. In the group fed with the mixture of Hungarian mulberry genotypes 20% of the larvae failed to spin.

The groups fed with reference sericultural *M. alba* varieties ‘Florio’ and ‘Morettiana’ had 10% losses, while the larvae fed with leaves of variety ‘Kokusou’ successfully spinned.

All larvae in groups fed with *M. alba* Bulgarian fruit genotype and *M. australis* leaves have successfully spinned. 5% losses were observed in the group fed with the *M. alba × rubra*. None of the larvae fed with *M. nigra* leaves spinned cocoons, however, food was offered until 12 days after the last moult.

#### 3.9.1. Molecular Detection of BmNPV by Real-Time PCR

The rt-PCR assays resulted in amplification signals (cT values at 17–21) in DNA extracts of diseased larvae. The nucleotide sequences of the amplification products have shown >99% similarity to an 81 bp long region of the polymerase gene region (between nt positions 47,965 and 48,045) of the BmNPV reference sequence (GenBank accession number: NC_001962). No amplification was detected in the negative control samples (healthy larvae).

Clinically healthy larvae were collected randomly from each experimental group before each moulting and spinning and were tested for BmNPV infection by real-time PCR: all real time-PCR assays in regularly sampled healthy larvae from each group were negative.

BmNPV DNA was detected in the dead or cachectic body of third and fourth instar larvae from the common box:1st day, third instar: 1 dead (liquid rotting) larva; cT mean 40.63, standard deviation (SD) 0.84;2nd day, third instar: 1 cachectic larva; cT mean 40.70, SD 1.49;1st day, fourth instar: 1 larva died during moulting, showed signs of Grasserie; cT mean 18.32, SD 0.17.In the fifth instar, BmNPV DNA was detected in three out of the 1340 larvae (0.22%). Positive larvae were ill or were found dead:1st day: Hu-BE 1264/2 (signs of Grasserie); cT mean 37.38, cT SD 0.69;end of 5th instar: Hu-ZA 2084.2 (moribund larva with black stripes), cT mean 20-09, SD 0.14;end of 5th instar: Hu-mix (dead larva, signs of Grasserie), CT mean 37.43, SD 0.92.

BmNPV DNA was not detected in any of the 39 larvae extracted from cocoons showing signs of failed spinning.

#### 3.9.2. Identification of Gut Microbiome and Potential Bacterial Pathogens by Metagenomic Studies

In the group of Slovenian genotypes mixture, approximately 102,489 different bacterial species were detected. In the group of Hungarian genotypes mixture, the number of detected species exceeded 106,000, while in the group reference genotypes mixture it exceeded 123,958. In group *M. australis* approximately 143,366, and in group *M. nigra* 144,551 different bacterial species were detected. Core bacterial microbiomes are shown in Figure 5.

While on the 5th day of fifth instar all larvae were still eating, the gut microbiomes differed considerably in the samples of larvae fed with mixtures of Slovenian, Hungarian, and reference genotypes of *Morus alba*, as well as the fruit varieties *M. australis* and *M. nigra*.

Major differences in silkworm gut bacterial microbiome in different groups on the level of bacterial classes and Chloroplast are the followings:(1)57% of representatives of Chloroplast (99% Streptophyta) were found by Slovenian genotypes and 47% by Hungarian genotypes, almost half less (22%) by reference genotypes, while *M. australis* and *M. nigra* had only 16% and 12%.(2)The range was quite opposite by class Betaproteobacteria (mainly representatives from order Burkholderiales): 44% by *M. nigra*, 39% by reference genotypes, 24% by Slovenian genotypes, 22% by Hungarian genotypes, and 18% by *M. australis*.(3)The representatives from Alphaproteobacteria (99% of representatives coming from order Rickettsiales) were in the following ranges: *M. nigra* and Hungarian genotypes 8%, Slovenian genotypes 6%, reference genotypes 3%, and *M. australis* 2% (representatives mainly from orders: Rhodobacterales 40%, Rhizobiales 24%, Sphingomonadales 12%, Rhodospirillales 12%, 9% Rickettsiales).(4)The representatives from class Gammaproteobacteria were in following ranges: Hungarian mulberry genotypes 4% (Pseudomonadales 3%, Enterobacteriaceae 1%, Xanthomonadales 0.1%), *M. nigra* 3% (Pseudomonadales 2%, Enterobacteriaceae 0.07%, Xanthomonadales 0.1%, Oceanospirillales 0.1%, Alteromonadales 0.1%), Slovenian genotypes 1% (Pseudomonadales 1%, Enterobacteriaceae 0.06%), reference genotypes 1% (Pseudomonadales 0.7%, Enterobacteriaceae 0.4%), *M. australis* 1% (Pseudomonadales 0.8%, Enterobacteriaceae 0.3%).(5)The representatives from class Actinobacteria were in the following ranges: reference genotypes 14%, Hungarian genotypes 9%, *M. australis* 9%, *M. nigra* 5%, and Slovenian genotypes 4%.(6)The representatives from class Bacilli were in the following ranges: Slovenian genotypes 4%, Hungarian genotypes 9%, reference genotypes 14%, *M. australis* 9%, and *M. nigra* 5%.(7)The representatives from class Clostridia were in the following ranges: *M. nigra* 3%, *M. australis* 2%, Slovenian genotypes 1%, Hungarian genotypes 1%, reference genotypes 0.8%.(8)The representatives from class Flavobacteriales were in the following ranges: *M. nigra* 3%, *M. australis* 2%, Slovenian genotypes 1%, Hungarian genotypes 1%, reference genotypes 0.8%.(9)The representatives from other classes were in the following ranges: Slovenian genotypes 4%, Hungarian genotypes 3%, reference genotypes 2.2%, *M. nigra* 2%, *M. australis* 1.2 %.

In comparison with other groups, Slovenian genotypes had the highest percentage of Chloroplast and lowest percentage of Gammaproteobacteria, Actinobacteria, Bacilli, Clostridia, and Flavobacteriale.

Hungarian genotypes had the highest percentages of Chloroplast, Alphaproteobacteria, and Gammaproteobacteria and the lowest of Bacilli, Clostridia, and Flavobacteriales.

In reference genotypes, the highest percentage of Actinobacteria and low percentage of Gammaproteobacteria, as well as Clostridia and Flavobacteriales, were described.

The group of *M. australis* had the lowest percentage of Chloroplast, Betaproteobacteria, Alphaproteobacteria, Gammaproteobacteria, and Flavobacteriales while the highest percentage of Bacilli of all groups.

The group of *M. nigra* had the highest percentage of Alphaproteobacteria and Clostridia, second highest of Betaproteobacteria, and second lowest percentage of Chloroplast.

Rikettsiales were the most representative order of class Alphaproteobacteria by all groups. The highest percentage of order Enterobacteriaceae was found by the Hungarian genotypes group, the lowest by *M. nigra*. The highest percentage of Lactobacillales was described by the group fed with *M. australis*.

The lowest percentage of family Microbacteriaceae was found by Slovenian genotypes and *M. nigra*, and the highest by reference genotypes and Hungarian genotypes. The highest percentage of family Micrococcaceae was found by reference genotypes and *M. australis*; the highest percentage of family Bifidobacteriaceae was found by *M. australis* and the lowest by Hungarian genotypes and reference genotypes groups. The lowest percentage of family Staphylococcaceae was found by Slovenian genotypes and Hungarian genotypes, highest by *M. australis*.

#### 3.9.3. Comparison of Selected Qualitative and Quantitative Hemogram Parameters in the Main Feeding Groups of Larvae

The highest total number of haemocytes was found in larvae fed with Hungarian mulberry genotypes (74.3 × 10^4^/mL; SE 5.8 × 10^4^/mL), followed by Slovenian genotypes (62.8 × 10^4^/mL; SE 7 × 10^4^/mL), reference genotypes (61.2 × 10^4^/mL; SE 2.6 × 10^4^/mL), *M. nigra* (58.3 × 10^4^/mL; SE 11.2 × 10^4^/mL) and *M. australis* (45.9 × 10^4^/mL; SE 6.6 × 10^4^/mL).

The qualitative analysis of the different haemocyte subsets revealed the dominance of granular cells, followed by plasmatocytes and small spheric cells (Figure 6).

## 4. Discussion

The main factors that contribute to the success of cocoon harvest are mulberry feed, environmental conditions, silkworm cultivation techniques, and silkworm strains [29]. Mulberry leaves are known as a rich source of proteins with unique amino acid composition and are highly palatable and digestible for herbivorous animals [6,30,31,32]. It has been previously confirmed that high protein content in leaves has a direct impact on the growth of larvae and cocoon production [28,33,34,35,36].

Mulberry leaves are further known by their unique composition of phenolic compounds, which were shown to have biological properties [6,37,38,39,40,41,42,43,44]. The identification of phenolics in mulberry leaves with UPLC-MS has been the subject of only a few recent studies so far [44,45].

The main phenolic compound in mulberry leaves was identified as chlorogenic acid followed by other hydroxycinnamic acids, including caffeoylquinic acid, p-coumaric acid derivatives, and p-coumaroylquinic acid derivatives [6,7,44,45,46,47]. The flavonol’s fraction mostly contains quercetin-3-O-rutinoside (rutin) as well as other quercetin and kaempferol glycosides [6,7,46,48,49,50,51,52].

It has been previously confirmed that chlorogenic acid enhances the rate of silkworm development. Furthermore, quercetin and kaempferol glycosides have been found to be transferred from the larval diet into the hemolymph and cocoons, where they act as UV shields and antimicrobial barriers and might therefore increase the survival rate of developing insects inside the cocoons [53,54,55,56,57,58]. However, when treated artificially, flavonoids may act as antinutrients by binding to amino acids and proteins as well as digestive gut enzymes they may reduce the nutrient value of mulberry leaves [50].

Based on mulberry inventory in Slovenia and Hungary, we established a collection of historical mulberry trees and screened their biochemical patterns regarding important primary (proteins and amino acids) metabolites and phenolics; we aimed for a definition of high-yielding and nutritive richer mulberry genotypes from the local Slovenian and Hungarian gene pool, which were included in the presented feeding experiment and compared with reference sericultural and fruit varieties. The analysis of study result data indicates possible influences of mulberry chemical parameters on silkworm development and production.

### 4.1. The Influence of Proteins and Phenolics on Larval, Cocoon and Raw Silk Parameters

The silkworm’s growth, cocoon, and raw silk quality can be affected by their feed sources to a great extent. The results of the screening of the metabolites in the leaves of Slovenian and Hungarian genotypes from their local origin were previously published by Urbanek Krajnc et al., and Šelih et al. [6,7]. These data served as the basis for the current research into the impact of feeding silkworms with selected mulberry genotypes out of the local gene pool. Based on biochemical analyses, we selected those genotypes of old local mulberries that proved to be the most favorable for silkworm rearing in terms of nutritional value and leaf yield.

The protein content of mulberry leaves ranges between 13 to 31% [29,51,59,60]. In the current experiment, the highest total protein content was determined in the Slovenian genotype SM 101.1 (239.42 mg/g DW), the lowest in the fruit variety Bulgarian accession (151.50 mg/g DW). The results further coincide with our previous study comprising local trees of the Gorizian region alone and Slovenian old mulberry genotypes sampled at the place of their origin with respect to pruning management and eco-geographical origin [6,7]. We confirmed a strong positive correlation between total proteins and raw silk length and weight. It has been previously confirmed that high protein content in leaves has a direct impact on the growth of larvae and cocoon production [29,61].

Mulberry leaves are also known to have high contents of phenolics, that in our varieties ranged between 11.26 and 21.51 mg/g DW. They were highest in the *M. alba* × *rubra* variety (21.51 mg/g DW) and lowest in SM 6 (11.26 mg/g DW). The analyzed concentrations were in accordance with those of Sánchez-Salcedo et al., who found phenolics in mulberry leaves in the range between 12.8 and 15.5 g GAE/g DW [62].

As previously reported, the predominant caffeoylquinic acid derivative was chlorogenic acid followed by 4-caffeoylquinic acid. In our feeding experiment, the average concentration of chlorogenic acid ranged between 4.00 and 18.05 mg/g DW, with the maximum content analyzed in *M. nigra* [6,7]. Amongst the Slovenian genotypes, the maximum content was found in genotype SE 5. Screening of phenolics on Slovenian genotypes sampled at the place of their origin revealed chlorogenic acid concentration in the range between 1.80 and 6.89 mg/g DW, whereas the maximum content was determined in genotype from the SM region. The concentrations are in accordance with other authors reporting the concentrations of chlorogenic acid in the range between 3 and 10 mg/g DW [43,49,62].

Generally, the predominant phenolic acids are known to have a stimulative effect on feeding, growth, and development [63]. Chlorogenic acid and other dihydroxybenzoid compounds isolated from mulberry leaves were found to be beneficial for silkworms’ growth and development. Chlorogenic acid is sensed by chemosensory organs in the mouthparts and stimulates feeding; hence, high concentrations of chlorogenic acid in mulberry leaves significantly promote feeding and correlate with growth parameters of silkworm larvae [64,65,66,67,68]. Furthermore, Yamagishi et al., identified an additional role of chlorogenic acid in the mid-gut lumen as a cue inducing the tachykinin-related peptide secretion from enteroendocrine cells [63]. These peptide hormones are known to modulate physiological processes such as the release of other hormones, secretion of digestive enzymes, gut motility, feeding behaviour, and energy homeostasis. Thus, the silkworm might use chlorogenic acid in differentially directed functions as a food marker in both the mouthparts and mid-gut.

The beneficial effect also correlates with increased silk production and quality. Naik et al., reported that supplementation of chlorogenic acid increased silk productivity and had a positive influence on different silk parameters, such as silk filament length and weight as well as silk protein fibration [69]. Our results did not show a correlative relationship of silkworm parameters with chlorogenic acid, but out of phenolic acids, there was a negative correlation between 5-coumaroylquinic acid and silkworm weight and p coumaroylquinic acid hexoside and the silk thread length.

In our study, the predominant flavonoids were quercetin and kaempferol glycosides. The main quercetin glycosides were rutin, quercetin malonyl-hexoside, and quercetin-3-glucoside (isoquercetin), whereas the predominant kaempferol glycoside was kaempferol acetyl-hexoside, that were generally recognized as the main flavonols in mulberry leaves [6,7,44,45,70,71]. The nutritional effect of flavonols on the growth and development of silkworm larvae and cocoon formation has been intensively studied [53,54,55,56,57,58,71,72,73,74]. In low concentrations, they have a beneficial effect on growth and development, whereas high doses might have an antinutrient effect [75].

Quercetin malonyl-hexoside has been recognized as the main quercetin glycoside with antioxidant activities [43,45,49,71,76,77,78,79]. In our sampled genotypes the mean concentrations ranged between 0.74 and 2.25 mg/g DW with the highest value in sericultural variety ‘Kokusou-20′. A high correlation was found between the length and weight of raw silk.

The second predominant flavonol was rutin, the maximum concentration was 4.31 mg/g DW determined in one Hungarian genotype (TO 1131). We found a medium correlation only with the struse of the raw silk. By reviewing the literature studying the effect of rutin on silkworm larvae, it was reported that rutin has no stimulative effect on the behaviour of silkworm larvae, although it stimulates feeding on many insects. Furthermore, a significant effect of rutin on the growth of silkworm larvae was not confirmed [74]. However, the authors were able to confirm that larvae can differentiate among quercetin glycosides of mulberry leaves. Quercetin-3-glucoside was recognized as a feeding stimulant but the rhamnose conjugate may deter feeding [74]. Furthermore, a positive effect of quercetin-3-glucoside on the growth and development of silkworm larvae was determined [66,74]. In the presented study, the trend towards a weak negative correlation was determined for silkworm weight and silk thread parameters.

The main kaempferol glycoside that was analyzed in leaves of local mulberries was identified as kaempferol acetyl-hexoside (0.94–4.42 mg/g DW), which reached the highest amount in *M. alba* × *rubra*. Other authors determined kaempferolhexoside in concentrations up to 0.75 mg/g DW, whereas kaempferol malonyl-hexoside was found in traces [71].

Interestingly, we found a negative correlation between kaempherol dirhamnosyl-hexoside and raw silk length and weight, whereas kaempherol rhamnosyl-hexoside correlated positively with struse. The studies on biologically active kaempferol derivatives in mulberry leaves are scarce, and it remains to be seen whether these derivatives may deter feeding.

### 4.2. The Influence of Macro- and Microelements on Larval, Cocoon and Raw Silk Parameters

Besides phenolics as bioactive compounds, minerals are among the important biochemical components of mulberry leaves, and they may have a high influence on silkworm, cocoon, and raw silk parameters. Previously, positive correlations of nitrogen, phosphorus, potassium, calcium, magnesium, and sulphur were obtained with larval, cocoon, and egg production parameters of mulberry silkworms [80,81]. Shifa et al. considered these macroelements as basic parameters for the evaluation of mulberry varieties for mulberry silkworms rearing in the future [81]. Based on this, we hypothesized that a high amount of these macroelements in mulberry leaves of selected mulberry genotypes will significantly contribute to silkworm and silk thread parameters.

Phosphorus (P) is an important major nutrient in the mulberry plant. It is a component of the complex nucleic acid structure of plants, which regulates protein synthesis. Therefore, it is very important in cell division and the development of new tissue. Phosphorus is also associated with complex energy transformations such as ATP [80,82]. An inadequate amount of P level affects the uptake of other nutritive elements in mulberry leaves for various other physiological activities, in turn, it hampers the growth and economic characteristics of silkworms [83,84]. The highest concentration of P was found in Slovenian genotype SE 5 (2.04 g/kg DW), which was superior in silk thread weight and thickness. When compared to other authors, Shifa et al., determined a minimum value of 1.11 g/kg in ‘Jimma coll’ and a maximum record of 3.22 g/kg in M-4 accession [81]. Similar to Shifa et al., we found no significant correlation between P and silk parameters or with silkworm weight [81].

Sulphur (S) is known to have an important role in the synthesis of proteins, oils, and vitamins [85]. It plays a vital role in the N metabolism and thus proper development of mulberry [86]. It is a constituent of S-containing amino acids, cysteine (contains 27% of S), and methionine (contains 21% of S). Methionine forms one of the ten essential amino acids for silk formation in silkworms. Cystine and cysteine are among the non-essential amino acids, the quantitative presence of which influences the formation of fibroin over sericin [87]. Deficiency of S level leads to low levels of S-containing amino acids, thus reducing protein synthesis. As a result, amino acids without S and amides of nitrate ions accumulate in the plant tissue and lead to a decrease in sugar as well as insoluble N (protein) in plants [86]. Similar to Ca, the highest content of S was found in the Hungarian genotype GMS 2532 (1260 mg/kg DW). The S concentrations of mulberry varieties analysed by Shifa et al., ranged from 0.15 g/kg in K-2 to 0.34 g/kg in M-4 accession [81]. In the presented experiment S ranged between a minimum value of 0.46 g/kg and a maximum value of 1.26 g/kg, which is up to four-fold higher than what was reported by Shifa et al. [81]. These could be due to the different soil conditions and because the mulberry gene bank is on the silicate geological basis of the southern slopes of Pohorje mountain (central alpine region) which might positively affect the uptake of several minerals (Zn, S, Fe), but negatively Ca, Mg, K, P [82]. Shifa et al., found S to have a significant positive correlation with larval weight, cocoon weight, and shell weight. In contrast to these authors, we did not find any correlation with measured parameters [81].

Potassium (K) plays important regulatory roles mostly in cell ion homeostasis, and stomatal conductance, and thus in maintaining water potential on the cell and whole plant level. Furthermore, it is known that the starch synthetase is activated by K. Thus, with inadequate K, the level of starch declines while soluble carbohydrates and N compounds accumulate. Therefore, it also plays a significant role in the high yield and quality of leaves [88]. It is also involved in the translocation of carbohydrates, protein metabolism, and pathogen tolerance in mulberry [80]. In the silkworm body, the strong alkalinity of the gastric juice originates from potassium and sodium compounds present in the haemolymph. The high alkaline condition of digestive fluid has strong germicidal power against pathogens. K is a unique element that contributes to the growth of silkworms to the maximum extent. In addition, K has a stimulating effect on protein synthesis including silk protein in the silk glands [89]. In the presented experiment, K was increased in fruit varieties, although insignificant. The highest concentration of K was present in Hungarian genotype BA 2225 (22.10 g/kg DW), which was also characterized by superior raw silk length, weight, and thickness as well as silk waste parameters. Shifa et al., reported the K contents of mulberry varieties with ranges from 11.35 g/kg in local varieties to 18.61 g/kg in M-4 accession [81].

Calcium (Ca), in the form of calcium pectate, is important for the cell wall structure in plant. Its deficiency causes incomplete cell division or mitosis, without the formation of a new cell wall resulting in multi-nuclear cells. Calcium is also important in activating certain enzymes and to acts as second messengers in cell signalling that coordinates certain cellular activities. Calcium acts as a detoxifying agent by neutralizing organic acids such as oxalic acid which helps in membrane stability and maintenance of chromosome structure, the activity of enzymes, and translocation of carbohydrates. It is also involved in the differential permeability of membranes [82]. Superior Ca contents were found in genotype GMS 2532 (20.00 g/kg DW). Shifa et al., reported calcium concentrations in the range from 13.45 mg/kg (local check) to 20.52 mg/kg (M-4), which is similar to our findings [81].

Micronutrients are needed in small quantities and they play a pivotal role in the enzymatic reactions and thus govern the growth, development, and yield of mulberries. Chloride (Cl) is involved in the hydrolysis of water in photosynthesis, the synthesis of starch, cellulose, and lignin. It influences cell homeostasis (water holding capacity) of plant tissues. It stimulates the activities of some enzymes [82]. Cl correlated with the weight of raw silk. The highest content of Cl was in the Hungarian genotype BE 1264.2.

Manganese (Mn) is essential for the synthesis of chlorophyll and the hydrolysis of water in photosynthesis, and its principal function is to activate some of the enzyme systems in plant physiology and regulation of Fe metabolism. In addition, it has a close relation with N metabolism, assimilation of carbohydrates, and formation of ascorbate. It is involved in redox processes and electron transport systems [82]. Similar to Fe, Mn has the potential to enhance larval development, filament length, cocoon weight, and yield [90]. Mn was the highest in Hungarian genotype BA 2225 (105 mg/kg DW).

Iron (Fe) is present in the chloroplast proteins and several enzymes. It plays a dominant role in protein metabolism and N fixation [82]. Fe has the potential to enhance larval (silkworm) development, filament length of a single cocoon, cocoon weight, and yield [90]. The altered Fe content in mulberry foliage resulted in reduced larval weight, cocoon weight, and silk filament length [89]. In the presented experiment, Fe correlated strongly with telette. The highest content of Fe was found in genotype GMS 2533.

Zinc (Zn) correlates strongly with silk filament length and pupal weight, whereas the excess Zn content in mulberry leaves leads to a reduction in cocoon yield [84,90]. In our experiment, the highest Zn content was found in Slovenian genotype SM6. However, no correlative relationship with Zn was detected in our analysis. We further found a negative correlation between Rb and silkworm weight and a positive correlation between Sr and telette.

In the presented experiment, significant differences between the mulberry groups in the contents of the main elements were determined for S, Mn, Fe, and Sr, which were based on the PCA analysis considered as important markers in the selection of mulberry feed source.

### 4.3. Correlations between Test Parameters in Multivariate Analysis

The foliar protein, phenolics, and mineral composition of mulberry varieties resulted in significant inter-relationship with larval, cocoon, and silk thread parameters when their leaves served as feeds. This relationship between leaf composition values and important silkworm and raw silk traits has been worked out through correlation and multivariate (PCA) analysis.

Function 1 clearly separated reference varieties, which were characterized by higher larval weight, which coincided with high values of chlorogenic acid, rutin, and other quercetin glycosides, from Slovenian and Hungarian varieties with higher in proteins, length, and weight of raw silk. The second discriminant function was positively associated with cocoon weight, length of raw silk, total phenolics, Mn, and K, and negatively with Ca and quercetin glucoside. Function 2 clearly separated Hungarian mulberries (by over 87%) from Slovenian genotypes.

Positive correlation of coumaroylquinic acid derivatives, certain flavonols, phosphorus, sulphur, Cl, Ca, Mn, Fe, Ni and Rb were obtained with larval, cocoon, and silk thread parameters, whereas caffeoylquinic acid derivatives affected only the length of raw silk. Therefore, it is likely that the levels of these bioactive compounds and elements in mulberry leaves are important feed markers (basis parameters) to the gains on important mulberry silkworm parameters when these leaves served as feed sources.

### 4.4. Development and Health Status of Larvae

The measurement of body weight gain was used within the experiment to monitor larval development in the different groups. Bodyweight was most successfully increasing in groups of larvae fed with reference varieties of mulberry trees, followed by fruit varieties closely together with the Hungarian genotypes. The slowest gain of bodyweight and lowest weight on day seven of the fifth instar were observed by larvae fed with Slovenian genotypes of mulberry trees. Nonetheless, larvae with lower body weights at this instar started spinning approximately one day later. The additional one-day eating mainly compensated for their backlogs and they have started spinning with approximately the same weight as the other ones.

A correlation was observed between larval body weight in the last days of fifth instar and the weight of the fresh cocoons; however, differences between the averages of sub-groups of cocoons within Slovenian, Hungarian, reference, and fruit variety groups were only 5% (2.14–2.25 g of average fresh cocoon weights). The highest proportion of heavier cocoons was produced in the Hungarian groups. Additionally, the best silk thread parameters (length, weight, and thickness) were also described by cocoons from Hungarian groups, while most cocoon values from reference sericultural varieties and Slovenian groups were positioned within intermediate values. The lowest values of cocoon weight and silk thread parameters were described by fruit groups. Larvae fed with leaves of *M. nigra* showed the worst possible performance, as none of them started spinning (despite rapid larval bodyweight gains).

The general health status of silkworm groups was sufficient, as more than 99% of larvae reached the spinning stage healthy and started to spin (i.e., only three larvae out of 1270 died). By some (about 30) of larvae in the last day or two before spinning some common, general, mild signs of disease were observed: inactivity, cessation of eating, laying on the side of the rearing box, transparent skin, yellow/ivory colour. However, differentiation between signs of disease and the physiological changes connected to preparation for spinning was not obvious.

Within larvae that failed spinning BmNPV DNA was not detected, and all other randomly sampled larvae were also negative. However, BmNPV DNA was detected already in a dead and in a cachectic/ill larva in the third instar, though with very low viral DNA content (cT values > 40). BmNPV DNA was also detected in a dead larva after the third moult and in three ill/dead larvae in the fifth instar. Two of these larvae contained high amounts of BmNPV (cT 18–20). These molecular data indicate that BmNPV infection was present in (some of the) the larvae from the beginning of their life. Silkworm rearing has never been performed in the place of the experiment, and there is no known alternative, wild insect hosts of BmNPV, so the infection could not come from the environment, fomites, or contaminated leaves. Nevertheless, the virus amount in the vast majority of the larvae remained under detectable levels, and the larvae stayed healthy.

Whole genome sequencing of silkworm body tissue and 16S rRNA gene sequencing of bacteria in guts did not reveal DNA of BmDNV or relevant amounts of facultatively pathogenic bacteria (e.g., *Enterococcus (Streptococcus) faecalis*, *E. (S.) faecium*, Staphylococcal species, *Serratia marcescens*); however, there were considerable differences in proportions of detected sequences on the level of bacterial classes and orders between the groups (detailed analysis will be described elsewhere). Attempts on the identification of probiotic components of *B. mori* gut microbiota were reviewed by Barretto et al.: studies indicated the impact of Lactobacillus, Enterococcus, and Bacillus spp. as major gut microbiota components [91]. Besides competitive and antimicrobial effects on enteric pathogens, the probiotic effects of Actinobacteria (e.g., Actinomycetales, Bifidobacteriaceae) and Betaproteobacteria (e.g., Burkholderiales) contribute to the digestion with enzymes (e.g., protease, amylase, and lipase production) [91]. In the experimental groups, the highest abundance of DNA sequences from Burkholderiales was detected, followed by Actinomycetales and Bacillales. Lactobacillales (including Streptococcaceae, Lactobacillaceae, and Enterococcaceae) were found in the lowest amounts. The relative amounts of these detected sequences were higher in the reference, *M. australis* and *M. nigra* groups, however, it was mainly attributed to the high amounts of chloroplast-related sequences in the groups of Slovenian and Hungarian varieties. When chloroplast DNA was excluded from the analysis, the relative abundance of bacterial groups with suspected probiotic effects was similar in the different groups. Although the trees were cultured in the same collection (i.e., in a common microbial environment), the lower relative amounts of Burkholderiales, and higher relative amounts of Bacillales, Lactobacillales, and Bifidobacteriaceae were detected in the gut microbiome of larvae fed with *M. australis*, compared to the other four groups. No clear connection was identified between the gut microbiome compositions, larval bodyweight gain, cocoon production, and quality in this study.

The mean total haemocyte counts (THC) in the different groups ranged between 45.9 and 74.3 × 104 cells/mL; which is considerably higher than THCs reported by Nematollahian et al. [92]. However, the haemocyte subsets (based on cell morphology) were found similar. No significant differences were found in the haemocyte subset rations between the different feeding groups. Neither was any correlation identified between THCs, haemocyte subset ratios and larval bodyweight gain, cocoon production, and quality in this study.

## 5. Conclusions

The results of the present investigation showed that mulberry varieties of local genetic origin as compared to reference sericultural and fruit varieties showed wide qualitative and quantitative variation in chemical traits with respect to proteins, phenolics, and minerals.

As a result, positive correlations of total proteins and kaempferol derivatives were obtained with silk thread parameters (i.e., length, weight). Coumaric acid correlated negatively with raw silk length, whereas 5-coumaroylquinic acid with larval weight. In addition, a positive correlation was found between Cl and raw silk weight, whereas Rb had a negative correlation with larval weight. Fe and Sr correlated positively with reeling waste.

Hence, the present study reveals that the selection of mulberry varieties out of the local gene pool for rearing silkworms based up on foliar protein, specific phenolics and mineral constituents of the mulberry varieties is very important to optimise larval development, cocoon production, and raw silk parameters. However, more research should be carried out to support the current findings in consideration of varying periods of leaf picking and nutrient analysis, pruning management, and field performances of mulberry varieties in different regions when using these varieties as feed sources. Although larvae fed with reference varieties were quickest reaching the final body weight, individual Hungarian genotypes (BA2151, SO 1013, TO1131) showed promising results as the mean larval weight was the highest. Furthermore, we were able to recognize Slovenian and Hungarian varieties which gave superior raw silk parameters. The lowest was obtained when larvae were fed with the reference varieties ‘Giazzola’ and some fruit genotypes. When analysing the reeling wastes, the highest was obtained when larvae were fed with Hungarian varieties.

Besides the above results, it is important to consider that some of the reference sericultural and fruit varieties of the mulberry trees are starting to develop earlier in the season than the Slovenian and Hungarian genotypes, which is a risk factor for increased losses due to early spring freezes, a relatively frequent climatic condition in these countries. So later spring development of local genotypes of Slovenian and Hungarian trees and the resistance to freezing can compensate for the bigger leaf yields of reference sericultural varieties in field conditions.

While Saxena et al., in sericulture worldwide reported annual losses of almost 20% of potential cocoon production, our experiment demonstrates that the sustainable production of quality silk cocoons is possible in Slovenia and Hungary providing the selection of superior local genotypes and suitable, locally adapted rearing technology is applied [93].

## Figures and Tables

**Figure 1 insects-13-00836-f001:**
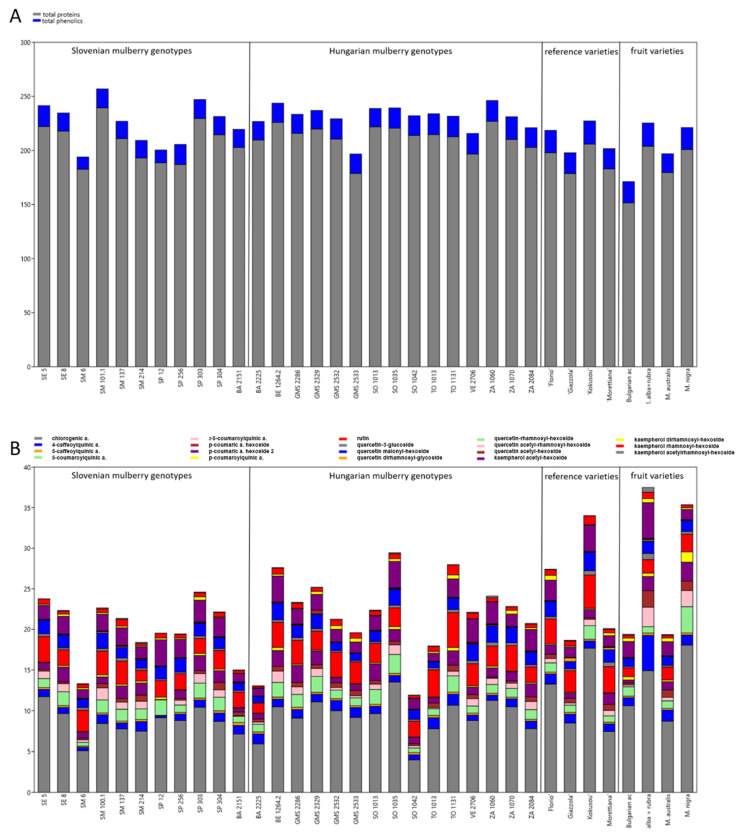
The mean concentrations of the (**A**) total proteins and total phenolics (mg/g DW), and (**B**) the concentrations of individual phenolics (mg/g DW) in leaves of Slovenian, Hungarian old mulberry genotypes, reference sericultural, and fruit varieties. For detailed data and statistics see Table 1 and Appendix A.

**Figure 2 insects-13-00836-f002:**
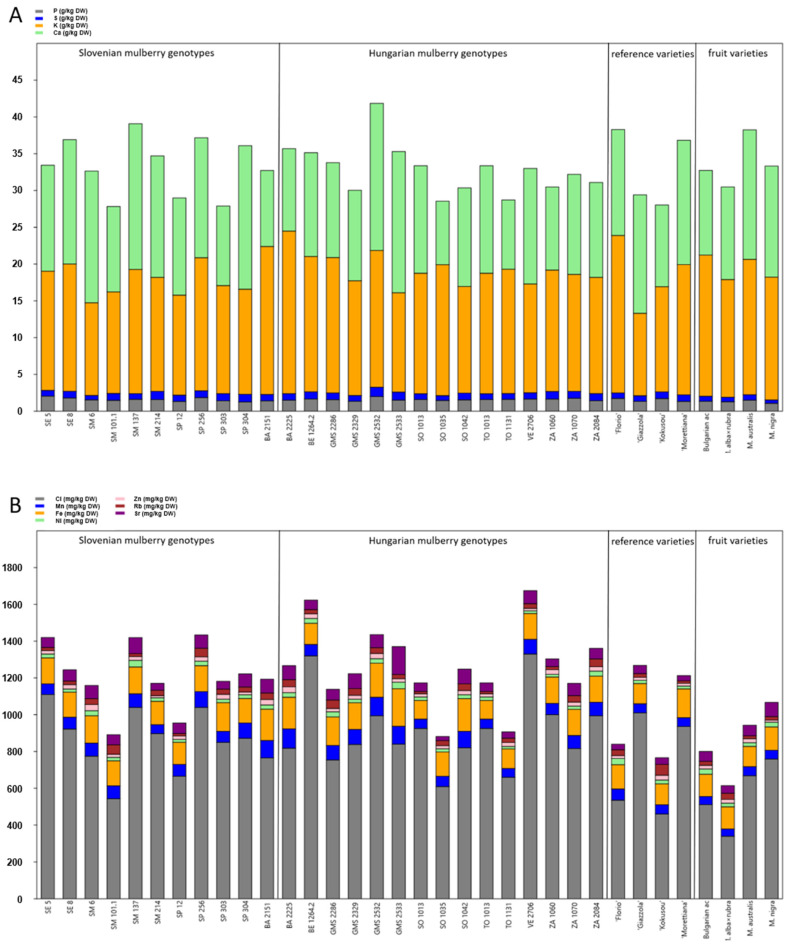
The mean concentrations of the (**A**) macroelements (mg/kg DW) and (**B**) microelements (mg/kg DW) in leaves of Slovenian, Hungarian old mulberry genotypes, reference sericultural and fruit varieties. For detailed data and statistics see Table 2 and Appendix A.

**Figure 3 insects-13-00836-f003:**
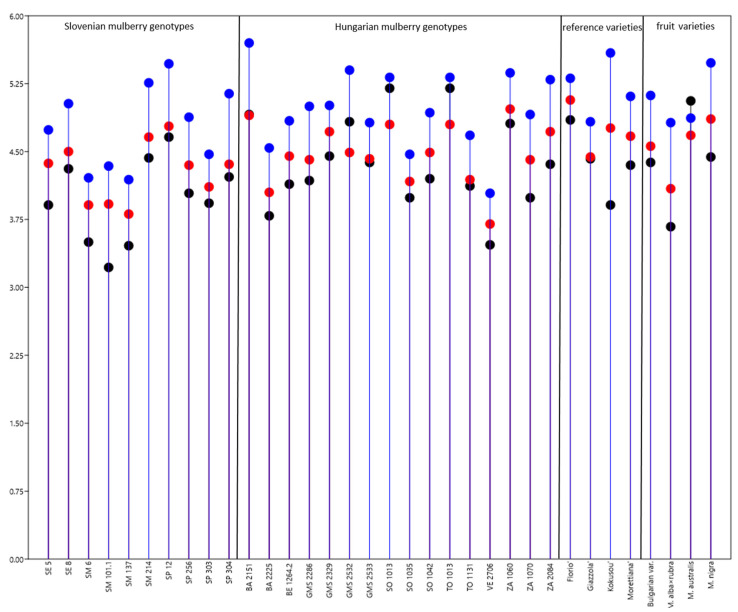
Mean weight of a larvae (±SD) on day 5 (black dots), 6 (red dots), 7 (blue dots) of the fifth instar when fed with Slovenian, Hungarian mulberry varieties, reference sericultural varieties, and fruit varieties. For detailed data and statistics see Table 3 and Appendix A.

**Figure 4 insects-13-00836-f004:**
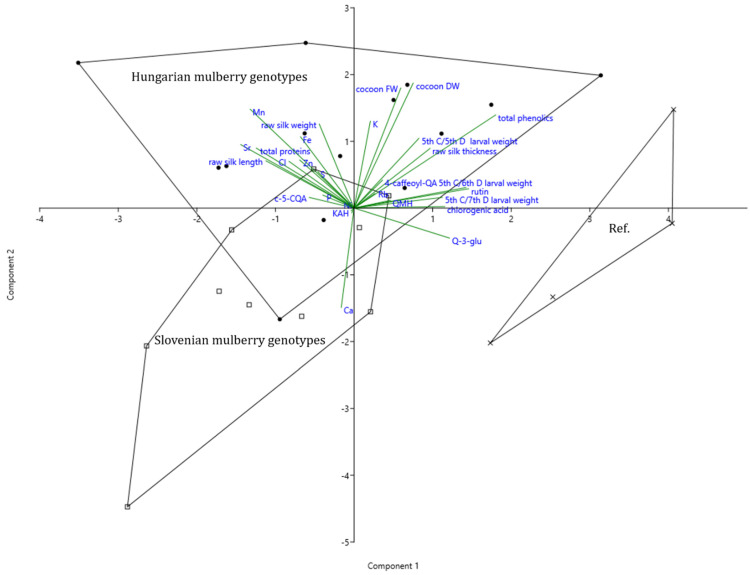
Principal component analysis of silkworm, cocoon, and raw silk parameters along with the chemical composition of leaves of different origins. The convex hulls delimit the space that includes the samples of mulberry trees from different Slovenian regions (squares), Hungarian regions (dots) compared to traditional high yielded sericultural varieties (Ref, crosses). * 4-caffeoyl-QA, 4-caffeoylquinnic acid; c-5-CQA, c-5-coumaroylquinnic a.; Q-3-glu, quercetin-3-glucoside; QMH, quercetin malonyl hex; KAH, kaemph acetyl-hexoside.

**Figure 5 insects-13-00836-f005:**
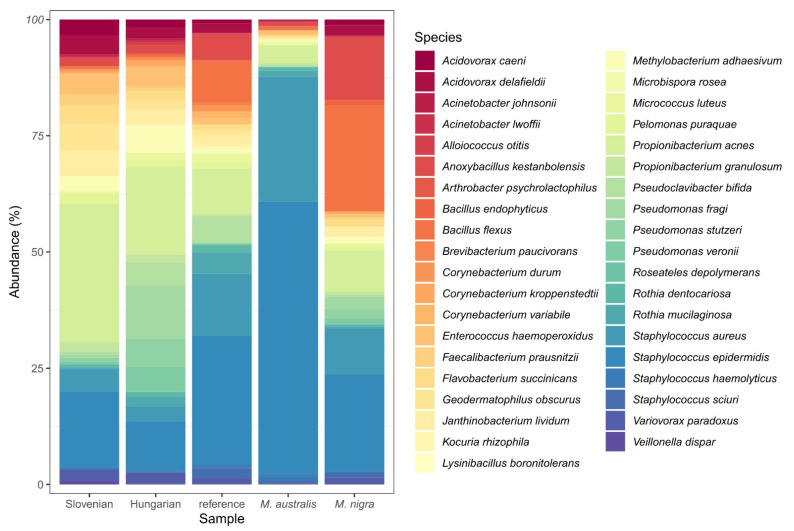
Species compositions of core gut bacterial microbiomes in the different experimental groups.

**Figure 6 insects-13-00836-f006:**
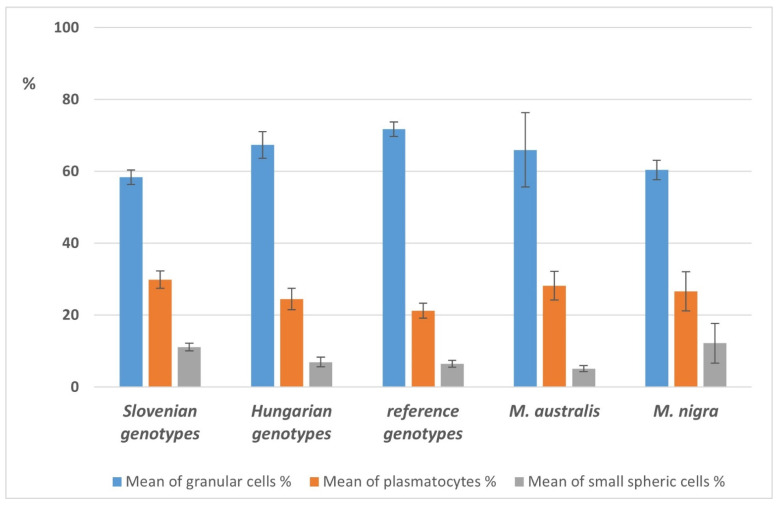
Differences in average haemocyte counts of larvae, fed with Slovenian, Hungarian, and reference sericultural genotypes and fruit varieties of mulberry trees.

**Table 1 insects-13-00836-t001:** Mean concentrations (±SD) of the total proteins, total phenolics, and predominant phenolics (mg/g DW) in Slovenian, Hungarian mulberry varieties, reference sericultural varieties, and fruit varieties. Different letters (a–c) indicate significant differences (*p* < 0.05) in the concentrations of specific compounds between the analysed groups, as determined by the post hoc Duncan test. *n*, number of repetitions.

	*n*	Total Proteins	Total PH	Chlorogenic a.	4-CQA	c-CQA	Rutin	Q-3-glu	QMH	KAH *
**Slovenian genotypes**	19	211.48 ± 18.51 a	17.03 ± 2.07 b	9.07 ± 1.66 c	0.9 2± 0.30 b	1.50 ± 0.32 a	2.32 ± 0.63 b	0.2 6 ± 0.06 b	1.51 ± 0.30 ab	2.15 ± 0.48 a
**Hungarian genotypes**	25	211.89 ± 13.77 a	18.19 ± 2.02 b	9.97 ± 1.61 bc	0.91 ± 0.25 b	1.52 ± 0.48 a	2.81 ± 0.64 a	0.31 ± 0.08 ab	1.63 ± 0.43 a	2.29 ± 0.65 a
**reference varieties**	4	191.38 ± 12.72 ab	19.95 ± 1.21 a	11.72 ± 4.36 ab	1.03 ± 0.14 b	1.12 ± 0.37 a	3.27 ± 0.57 a	0.40 ± 0.15 a	1.63 ± 0.53 a	2.09 ± 0.87 a
**fruit varieties**	4	183.95 ± 24.20 b	20.07 ± 2.03 a	13.43 ± 3.61 a	2.47 ± 1.64 a	1.53 ± 1.07 a	1.63 ± 0.40 c	0.41 ± 0.26 a	1.26 ± 0.24 b	2.35 ± 1.31 a

* t PH, total phenolics; 4-CQA, 4-caffeoylquinnic acid; c-CQA, c-5-coumaroyilquinnic acid; Q-3-glu, quercetin-3-glucoside; QMH, quercetin-malonyl-hexoside; KAH, kaempherol-acetyl-hexoside.

**Table 2 insects-13-00836-t002:** Mean concentrations (±SD) of the macroelements (g/kg DW) and the predominant microelements (mg/kg DW) in mulberry leaves in Slovenian, Hungarian mulberry varieties, reference sericultural varieties, and fruit varieties. Different letters (a, b) indicate significant differences (*p* < 0.05), which were determined using the post hoc Duncan test. *n*, number of repetitions.

	*n*	P	S	K	Ca	Cl	
**Slovenian genotypes**	19	1.75 ± 0.75	0.88 ± 0.23 a	15.92 ± 3.59	17.14 ± 4.90	788.16 ± 36.26	
**Hungarian genotypes**	25	1.42 ± 0.32	0.83 ± 0.19 ab	16.47 ± 3.16	14.03 ± 3.47	953.20 ± 34.69	
**reference varieties**	4	1.52 ± 0.23	0.84 ± 0.80 ab	16.15 ± 4.39	14.63 ± 2.57	735.25 ± 27.79	
**fruit varieties**	4	1.30 ± 0.18	0.62 ± 0.12 b	17.58 ± 1.48	14.20 ± 2.72	569.25 ± 18.45	
		**Mn**	**Fe**	**Ni**	**Zn**	**Rb**	**Sr**
**Slovenian genotypes**	19	66.04 ± 14.58 a	147.42 ± 26.27 ab	28.35 ± 11.88	23.91 ± 11.34	36.23 ± 28.72	54.02 ± 25.16 ab
**Hungarian genotypes**	25	67.32 ± 21.29 a	154.36 ± 29.89 a	24.19 ± 9.40	19.98 ± 5.66	26.75 ± 9.73	72.01 ± 31.40 a
**reference varieties**	4	52.03 ± 6.16 ab	128.25 ± 21.42 ab	21.68 ± 8.06	18.20 ± 5.32	31.20 ± 20.27	35.10 ± 7.16 b
**fruit varieties**	4	45.38 ± 4.22 b	119.50 ± 7.05 b	23.65 ± 4.20	18.40 ± 4.59	22.40 ± 7.73	57.53 ± 15.07 ab

**Table 3 insects-13-00836-t003:** Mean weight (±SD) of a larvae and cocoon fresh weight (FW, g) when fed with Slovenian, Hungarian mulberry varieties, reference sericultural varieties, and fruit varieties. Different letters (a, b) indicate significant differences (*p* < 0.05), which were determined using the post hoc Duncan test. FW, fresh weight. *n*, number of repetitions.

	*n*	5th Instar/5th D	5th Instar/6th D	5th Instar/7th D	Cocoon FW
**Slovenian mulberry genotypes**	19	3.97 ± 0.46 b	4.28 ± 0.33 b	4.77 ± 0.46 b	2.14 ± 0.12 a
**Hungarian mulberry genotypes**	25	4.32 ± 0.46 ab	4.46 ± 0.34 b	4.96 ± 0.43 b	2.26 ± 0.09 a
**reference sericultural varieties**	4	4.54 ± 0.38 a	4.85 ± 0.23 a	5.36 ± 0.36 a	2.25 ± 0.14 a
**fruit varieties**	4	4.39 ± 0.57 ab	4.55 ± 0.33 ab	5.07 ± 0.30 ab	2.20 ± 0.14 a

**Table 4 insects-13-00836-t004:** Mean (±SD) raw silk and reeling waste parameters (g) of Slovenian, Hungarian mulberry varieties, reference sericultural varieties, and fruit varieties. Different letters (a, b) indicate significant differences (*p* < 0.05), which were determined using the post hoc Duncan test. *n*, number of repetitions.

	*n*	Silk Thread Parameters			Reeling Wastes	
		Length	Weight	Thickness	Struse	Telette
**Slovenian genotypes**	95	1419.09 ± 64.14 a	0.43 ± 0.03 a	2.72 ± 0.13 a	0.036 ± 0.013 b	0.017 ± 0.009 a
**Hungarian genotypes**	211	1429.18 ± 94.82 a	0.44 ± 0.03 a	2.82 ± 0.15 a	0.047 ± 0.022 a	0.021 ± 0.019 a
**reference varieties**	70	1363.20 ± 135.94 a	0.43 ± 0.02 a	2.84 ± 0.19 a	0.043 ± 0.016 ab	0.017 ± 0.016 a
**fruit varieties**	30	1244.80 ± 44.03 b	0.37 ± 0.03 b	2.73 ± 0.13 a	0.040 ± 0.020 b	0.022 ± 0.016 a

**Table 5 insects-13-00836-t005:** Pearson’s correlation coefficient between silkworm, cocoon, and raw silk parameters in relation to the main phenolics (*n* = 32) in mulberry leaves listed according to their retention times on HPLC.

	Silkworm Weight (Larvae/g)				Cocoon Weight		Silk Thread		Reeling Waste		
Correlations	5th C/5th D	6th D	7th D	Mean 3d Weight	Fresh	Dry	Length	Weight	Thickness	Struse	Telette
**5th C/5th D**	1	00.856**	0.764 **	0.934 **	0.777 **	0.766 **	−0.11	0.191	0.438 *	0.234	0.121
**5th C/6th D**	0.856 **	1	0.907 **	0.967 **	0.713 **	0.715 **	−0.088	0.21	0.452 **	0.18	0.066
**5th C/7th D**	0.764 **	0.907 **	1	0.935 **	0.607 **	0.618 **	−0.032	0.166	0.298	0.226	0.133
**mean 3d weight**	0.934 **	0.967 **	0.935 **	1	0.744 **	0.745 **	−0.082	0.199	0.418 *	0.229	0.116
**cocoon FW**	0.777 **	0.713 **	0.607 **	0.744 **	1	0.992 **	0.024	0.376 *	0.562 **	0.193	0.081
**cocoon DW**	0.766 **	0.715 **	0.618 **	0.745 **	0.992 **	1	0.016	0.378 *	0.583 **	0.189	0.076
**length**	−0.11	−0.088	−0.032	−0.082	0.024	0.016	1	0.785 **	−0.075	−00.368 *	−0.183
**weight**	0.191	0.21	0.166	0.199	0.376 *	0.378 *	0.785 **	1	0.555 **	−00.455 **	−0.07
**thickness**	0.438 *	0.452 **	0.298	0.418 *	0.562 **	0.583 **	−0.075	0.555 **	1	−0.269	0.099
**struse**	0.234	0.18	0.226	0.229	0.193	0.189	−0.368 *	−0.455 **	−0.269	1	0.176
**telette**	0.121	0.066	0.133	0.116	0.081	0.076	−0.183	−0.07	0.099	0.176	1
**total proteins**	−0.198	−0.139	−0.116	−0.164	0.083	0.073	0.448 **	0.495 **	0.175	−0.024	−0.262
**total phenolics**	−0.001	0.131	0.137	0.087	0.096	0.108	−0.197	−0.12	0.069	0.191	0.13
**chlorogenic a.**	0.02	0.202	0.228	0.148	0.205	0.205	−0.198	−0.068	0.165	0.211	0.15
**4-caffeoyl-QA**	−0.092	−0.12	0.006	−0.071	−0.125	−0.136	−0.329	−0.312	−0.098	0.211	0.246
**5-caffeoyl-QA**	−0.031	−0.068	−0.002	−0.034	0.004	0.019	0.109	0.079	−0.023	0.046	−0.15
**total caffeoyl-QA**	−0.003	0.154	0.206	0.117	0.157	0.155	−0.248	−0.127	0.126	0.236	0.186
**c-5-CQA**	0.023	0.034	0.037	0.033	0.231	0.225	0.21	0.272	0.176	0.034	−0.167
**t-5-CQA**	−0.451 **	−0.328	−0.281	−0.382	−0.265	−0.28	−0.01	0.002	0.003	−0.1	−0.092
**p-CAH**	−0.048	−0.15	−0.059	−0.084	−0.123	−0.13	−0.378 *	−0.322	−0.045	0.158	0.196
**p-CAH2**	−0.044	−0.057	−0.1	−0.07	0.028	0.008	0.15	0.168	0.066	0.05	−0.281
**p-CQA**	−0.147	−0.181	−0.154	−0.168	−0.094	−0.09	−0.283	−0.313	−0.097	0.121	−0.034
**total coumaroyl-QA**	−0.183	−0.171	−0.146	−0.177	−0.04	−0.055	0.007	0.05	0.065	0.044	−0.145
**rutin**	−0.179	−0.054	−0.05	−0.108	0.081	0.101	−0.02	−0.041	−0.033	0.364 *	−0.018
**Q-3-glu**	−0.326	−0.105	−0.056	−0.185	−0.163	−0.153	−0.247	−0.342	−0.225	0.298	0.033
**QMH**	−0.268	0.02	−0.047	−0.122	0.098	0.11	0.085	0.177	0.193	−0.029	−0.161
**Q-diR-gly**	−0.082	−0.098	−0.132	−0.109	−0.155	−0.128	−0.209	−0.228	−0.088	0.086	0.011
**QRH**	−0.299	−0.301	−0.235	−0.295	−0.25	−0.27	−0.16	−0.304	−0.303	0.127	0.155
**Q-acetyl-RH**	−0.207	−0.185	−0.084	−0.168	−0.044	−0.065	−0.082	−0.125	−0.1	0.046	−0.047
**QAH**	0.009	0.135	0.006	0.045	0.284	0.275	−0.132	0.054	0.264	0.189	−0.12
**total quercetin-gly**	−0.27	−0.044	−0.071	−0.15	0.077	0.097	−0.037	−0.018	0.033	0.298	−0.077
**KAH**	−0.197	−0.071	−0.007	−0.105	−0.043	−0.044	−0.077	−0.028	0.066	0.086	0.054
**K-diRH**	0.126	0.054	−0.044	0.052	0.05	0.01	−0.525 **	−0.407 *	0.042	0.284	0.231
**KRH**	−0.199	−0.175	−0.075	−0.16	0.003	0.012	−0.152	−0.204	−0.125	0.482 **	0.106
**K-acetyl-RH**	−0.192	−0.175	−0.091	−0.163	−0.216	−0.23	−0.247	−0.324	−0.22	0.163	0.167
**total K-gly. deriv. ***	−0.194	−0.101	−0.038	−0.123	−0.046	−0.053	−0.202	−0.162	0.011	0.231	0.118

** Correlation is significant at the 0.01 level (2-tailed); * Correlation is significant at the 0.05 level (2-tailed). FW, fresh weight; DW, dry weight. * 4-caffeoyl-QA, 4-caffeoylquinic acid; 5-caffeoyl-QA, 5-caffeoylquinic acid; 5-total CQA, total caffeoylquinic acid derivatives; c-5-CQA, c-5-coumaroylquinic acid; t-5-CQA, t-5-coumaroylquinic acid; p-CAH, p-coumaric acid hexoside; p-CAH2, p-coumaric acid hexoside 2; p-CQA, p-coumaroylquinic acid; total CQA, total coumaroylquinic acid derivatives; Q-3-glu, quercetin-3-glucoside; QMH, quercetin malonyl-hexoside; Q-diR-gly, quercetin dirhamnosyl-glycoside; QRH, quercetin rhamnosyl-hexoside; Q-acetyl-RH, quercetin acetyl-rhamnosyl hexoside; QAH.

**Table 6 insects-13-00836-t006:** Pearson’s correlation coefficient between silkworm, cocoon, and raw silk parameters in relation to the macro- and microelements (*n* = 32) in **mulberry leaves**.

	Silkworm Weight (Larvae/g)				Cocoon Weight		Silk Thread Parameters			Reeling Waste	
Correlations	5th C/5th D	5th C/6th D	5th C/7th D	3d Weight	Fresh	Dry	Length	Weight	Thickness	Struse	Telette
**5th C/5th D**	1	0.856 **	0.764 **	0.934 **	0.777 **	0.766 **	−0.11	0.191	0.438 *	0.234	0.121
**5th C/6th D**	0.856 **	1	0.907 **	0.967 **	0.713 **	0.715 **	−0.088	0.21	0.452 **	0.18	0.066
**5th C/7th D**	0.764 **	0.907 **	1	0.935 **	0.607 **	0.618 **	−0.032	0.166	0.298	0.226	0.133
**mean 3d weight**	0.934 **	0.967 **	0.935 **	1	0.744 **	0.745 **	−0.082	0.199	0.418 *	0.229	0.116
**cocoon FW**	0.777 **	0.713 **	0.607 **	0.744 **	1	0.992 **	0.024	0.376 *	0.562 **	0.193	0.081
**cocoon DW**	0.766 **	0.715 **	0.618 **	0.745 **	0.992 **	1	0.016	0.378 *	0.583 **	0.189	0.076
**length**	−0.11	−0.088	−0.032	−0.082	0.024	0.016	1	0.785 **	−0.075	−0.368 *	−0.183
**weight**	0.191	0.21	0.166	0.199	0.376 *	0.378 *	0.785 **	1	0.555 **	−0.455 **	−0.07
**thickness**	0.438 *	0.452 **	0.298	0.418 *	0.562 **	0.583 **	−0.075	0.555 **	1	−0.269	0.099
**struse**	0.234	0.18	0.226	0.229	0.193	0.189	−0.368 *	−0.455 **	−0.269	1	0.176
**telette**	0.121	0.066	0.133	0.116	0.081	0.076	−0.183	−0.07	0.099	0.176	1
**P**	−0.016	0.001	−0.001	−0.006	0.144	0.129	0.269	0.261	0.052	−0.068	−0.097
**S**	0.207	0.212	0.347	0.271	0.075	0.08	0.228	0.287	0.135	−0.017	0.331
**K**	0.288	0.236	0.208	0.262	0.282	0.296	0.178	0.234	0.176	−0.14	−0.155
**Ca**	0.074	−0.108	−0.086	−0.032	−0.103	−0.141	−0.16	−0.1	0.021	−0.108	0.166
**Cl**	−0.003	−0.136	−0.194	−0.11	0.042	0.063	0.237	0.352 *	0.249	−0.211	−0.022
**Mn**	−0.058	−0.152	−0.05	−0.086	−0.035	−0.041	0.334	0.324	0.058	−0.106	0.226
**Fe**	−0.074	−0.115	−0.062	−0.086	−0.184	−0.202	0.167	0.113	−0.062	−0.019	0.434 *
**Ni**	−0.035	−0.027	−0.049	−0.04	−0.161	−0.211	0.027	0.012	−0.02	−0.151	0.185
**Zn**	−0.052	−0.048	0.025	−0.027	−0.139	−0.126	0.081	0.037	−0.032	−0.105	−0.016
**Rb**	−0.367	−0.094	0.056	−0.159	−0.276	−0.267	0.26	0.097	−0.185	−0.119	−0.16
**Sr**	−0.083	−0.205	−0.168	−0.154	−0.083	−0.123	−0.003	0.005	−0.024	−0.042	0.533 **

** Correlation is significant at the 0.01 level (2-tailed); * Correlation is significant at the 0.05 level (2-tailed); FW, fresh weight; DW, dry weight.

## Data Availability

Nucleotide sequence data were deposited in the NCBI SRA database under the BioProject accession number PRJNA879405. Raw data of chemical and microbiological analyses will be provided by the authors upon request.

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
