# Peer review of "The Effect of Feeding with Central European Local Mulberry Genotypes on the Development and Health Status of Silkworms and Quality Parameters of Raw Silk"

_insects, 2022, doi:10.3390/insects13090836_

Round 1
Reviewer 1 Report
The article has both scientific and practical importance. One of the most important things in order to revive the sericultural activities in Hungary and Slovenia is to have available suitable mulberry varieties, meeting the nutritional requirements of the present silkworm polyhybrids. Even though the sericulture was well developed in those countries in the past, the silkworm breeds reared at that time were quite different from the present day strains. For example in 18th and 19th century in Europe exclusively univoltine silkworm breeds were grown, while the present strains, used commercially have mostly bivoltine and even polyvoltine blood in their genetics. Therefore the present article deals with a very important problem, namely to investigate the suitability of the locally available mulberry varieties for feeding present silkworm polyhybrids, in comparison with the feeding with well known Italian and Japanese highly productive varieties and also with some mulberry fruit varieties.
The results, obtained have high scientific values such as:
- It is proved that mulberry varieties of local genetic origin as compared to reference sericultural and fruit varieties showed wide qualitative and quantitative variation in chemical traits with respect to proteins, phenolics and minerals.
- It was detected that different traits of larvae, cocoons and raw silk have significantly different relationships with various biochemical constituents of mulberry leaves.
- The present study reveals that the selection of mulberry varieties out of the local gene pool for rearing silkworms based up on foliar protein, specific phenolics and mineral constituents of the mulberry varieties is very important to get improvements in larval development, cocoon production and raw silk parameters.
Recommendation: As the authors intend to continue their research in this field they may wish to use also the gravimetric method, described by Waldbauer (1964), Horie at al. (1976), Horie and Watanabe (1983) in order to detect the food ingested, digested, food ingestibility and digestibility and efficiency of food conversion by the silkworm larvae.
Author Response
Dear Reviewer,
Thank you very much for the positive evaluation and concise summary of the work, and also many thanks for your valuable advice. We do intend to continue our studies and the mentioned gravimetric method would indeed provide useful additional data on the ingestibility and digestibility of mulberry leaves, as well as the efficiency of food conversion, which is an important aspect, particularly for larger scale rearings.
Reviewer 2 Report
General comment. The nutritional information of the plants should be better specified and may lead to more information. For example, is this information obtained on a fresh or dry basis? Has the dry matter been estimated? Has fat been estimated? Or more nutritional components? Ideally, a Table with all the nutritional information would be made.
The tables and figures are hard to see, please check their format.
Tables and figures must be self-exploitative.
Table 1: there are times that they put the letter c, however, it does not appear in the title of the Table.
Are they mean or lsmeans?
Are the differences by rows or by columns? Everything must be well written.
DW what does it mean? Everything must be well explained.
How do you think that such a low n (n=4) can affect the interpretation of the results?
Table 2. Cannot read. Review comments Table 1.
Table 3. The title does not understand the definition used for the different groups, please rewrite it.
Table 5. Very small numbers, not read correctly.
Figure 1. Are they means or lsmeans? The legend does not display correctly. Definition of D.W.
What does new Figure 1 and Figure 2 contribute with respect to Tables 1-3?
It seems in the abstract that the data with different protein levels are your results, and yet that is a cause and not a consequence of the work. I think it should be clear.
The conclusions are too long, please, make a correct summary of them and put the most important
Author Response
Dear Reviewer,
Thank you very much for the careful review of our manuscript, constructive comments and recommendations.
“General comment: The nutritional information of the plants should be better specified and may lead to more information. For example, is this information obtained on a fresh or dry basis? Has the dry matter been estimated? Has fat been estimated? Or more nutritional components? Ideally, a Table with all the nutritional information would be made.”
Our response: All nutrients were analysed on a dry weight basis. Dry matter was determined in range between 40.5-44.46% dependent on the genotype. Fat and other nutrients were not evaluated as we focused mainly on phenolic acids, which are known to have a stimulatory effect, quercetin and kaempherol glycosides, some of which have hardly been studied in relation to silkworm larvae. Sugars were not assessed as they can be influenced by various environmental factors.
“Comment: The tables and figures are hard to see, please check their format.
Tables and figures must be self-exploitative.”
Our response: The tables are formatted as recommended in the ‘Instructions for authors’ (Palatino Linotype 10), but uploaded in the MS Word (.docx) version as picture due to difficulties with pasting the MS Excel table. We used landscape page format for Table 1 and 2 in order to increase the size of the table. We ask the editorial office support for formatting of the tables and include them additionally as Excel version in recommended format (Palatino Linotype 10).
“Comment: Table 1: there are times that they put the letter c, however, it does not appear in the title of the Table.”
Our response: Letters have been corrected.
“Comment: Are they mean or lsmeans?”
Our response: We referred to observed means as average values. We explained this in the Materials and methods section (l. 353-354).
“Comment: Are the differences by rows or by columns? Everything must be well written.”
Our response: The differences are by columns within the certain compound.
- 437-439: “Mean concentrations (± SD) of the total proteins, total phenolics and predominant phenolics (mg/g DW) in mulberries in association with their origin. Different letters (a-c) indicate significant differences (p < 0.05) in the concentrations of specific compound between the analysed groups, as determined by the post hoc Duncan test.”
“Comment: DW what does it mean? Everything must be well explained.”
Our response: DW = dry weight. It is abbreviated in Materials and methods.
- 171-172: “The concentrations of the analyzed nutrients are calculated on a dry weight (DW) basis.”
“Comment: How do you think that such a low n (n=4) can affect the interpretation of the results?”
Our response: The n is referred to number of genotypes. For the statistical evaluation raw data were used. Each genotype was sampled on two sampling dates (16th and 23rd June 2021) during the feeding experiments in which three to five trees were sampled each and use as one sample, which was analysed in duplicates. It is explained in the section Materials and methods.
- 352-356: “The results of biochemical analyses of mulberry leaves are mean (average) values (± SD) of the analyses two sampling dates (16th and 23rd June 2021) during the feeding experiments in which three to five trees were sampled each and use as one sample. Measurements were performed at least four times for each sample and in duplicate.”
“Comment: Table 2. Cannot read. Review comments Table 1.”
Our response: The table was formatted as recommended in ‘Instructions for authors’.
“Comment: Table 3. The title does not understand the definition used for the different groups, please rewrite it.”
Our response: Title was rewritten.
“Table 3: Mean weight (± SD) of a larvae and cocoon fresh weight (FW, g) when fed with Slovenian, Hungarian mulberry varieties, reference sericultural varieties and fruit varieties. Different letters (a, b) indicate significant differences (p < 0.05), which were determined using the post hoc Duncan test.”
“Comment: Table 5. Very small numbers, not read correctly.”
Our response: Both Table 5 and 6 were formatted according to the ‘Instructions for authors’.
“Comment: Figure 1. Are they means or lsmeans? The legend does not display correctly. Definition of D.W.”
Our response: We referred to observed means as average values. We explained it in the Materials and methods section. DW = dry weight. It is abbreviated in Materials and methods section (l. 171-172).
“Comment: What does new Figure 1 and Figure 2 contribute with respect to Tables 1-3?”
Our response: In Figure 1 and 2 data are presented as stacked graphs represented by the main individually phenolic compounds/elements for each individual genotype out of the Slovenian, Hungarian mulberry varieties, reference sericultural varieties and fruit varieties. In Suppl. Tables 2 all analysed individual phenolics are shown and the main compounds are statistically evaluated among genotypes. Furthermore, the total values of phenolic acids, quercetin and kaempherol glycosides are presented. Table 1 and 2 represents the descriptive statistic of among the genotype groups (Slovenian, Hungarian mulberry varieties, reference sericultural varieties and fruit varieties).
“Comment: It seems in the abstract that the data with different protein levels are your results, and yet that is a cause and not a consequence of the work. I think it should be clear.”
Our response: The results, summary and abstract were checked for consistency. In the results, the nutrients were explained in more detail, while in the summary we only focused on the most important differences and were not able to describe them in detail due to the limited number of characters.
“Comment: The conclusions are too long, please, make a correct summary of them and put the most important.”
Our response: The conclusion was shortened and partly rewritten.
- 1249-1282: “The results of the present investigation showed that mulberry varieties of local genetic origin as compared to reference sericultural and fruit varieties showed wide qualitative and quantitative variation in chemical traits with respect to proteins, phenolics and minerals.
As a result, positive correlations of total proteins and kaempferol derivatives were obtained with silk thread parameters (i.e., length, weight). Coumaric acid correlated negatively with raw silk length, whereas 5-coumaroylquinic acid with larval weight. In addition, a positive correlation was found between Cl and raw silk weight, whereas Rb had a negative correlation with larval weight. Fe and Sr correlated positively with reeling waste.
Hence, the present study reveals that the selection of mulberry varieties out of the local gene pool for rearing silkworms based up on foliar protein, specific phenolics and mineral constituents of the mulberry varieties is very important to optimise larval development, cocoon production and raw silk parameters. However, more research should be carried out to support the current findings in consideration with varying period of leaf picking and nutrient analysis, pruning management, field performances of mulberry varieties in different regions when using these varieties as feed sources. Although larvae fed with reference varieties were quickest reaching the final bodyweight, individual Hungarian genotypes (BA2151, SO 1013, TO1131) showed promising results as the mean larval weight was the highest. Furthermore, we were able to recognize Slovenian and Hungarian varieties which gave superior raw silk parameters. The lowest were obtained when larvae were fed with the reference varieties ‘Giazzola’ and some fruit genotypes. When analysing the reeling wastes, the highest were obtained when larvae were fed with Hungarian varieties.
Beside the above results, it is important to consider that some of reference sericultural and fruit varieties of the mulberry trees are starting to develop earlier in the season than the Slovenian and Hungarian genotypes, which is a risk factor for increased losses due to early spring freezes, a relatively frequent climatic condition in these countries. So later spring development of local genotypes of Slovenian and Hungarian trees and the resistance to freezing can compensate the bigger leaf yields of reference sericultural varieties in field conditions.
While Saxena et al. in sericulture worldwide reported annual losses of almost 20% of potential cocoon production, our experiment demonstrates that the sustainable production of quality silk cocoons is possible in Slovenia and Hungary providing the selection of superior local genotypes and suitable, locally adapted rearing technology is applied [93].”
Round 2
Reviewer 2 Report
Thank you for having taken my suggestions into account. However, although it is explained in the materials and methods, any abbreviation must be reflected in each table and/or figure used. For everything else, perfect.